# OpenReview forum: "AlphaDPO: Adaptive Reward Margin for Direct Preference Optimization"
_ICML.cc/2025/Conference — ICML 2025 poster_

### Official Review · Reviewer_TEY3 · 2025-02-28

**Overall Recommendation:** 1

**Summary:**

This paper proposes AlphaDPO which is a direct preference optimization method with a data-dependent margin. The authors first observe that the DPO objective can be looked at as making the likelihood of the chosen response to be greater than the losing response, with a margin set as the difference between the likelihood of the responses under the reference model. The authors claim that such margin might introduce error in scenarios where the reference model is not calibrated. Therefore, they propose a margin that is the likelihood ratio of the responses under the language model and the reference model. They show how this proposal is theoretically related to SimPO and TDPO. They further empirically compare AlphaDPO with SimPO and several other DPO variants on AlpacaEval with Llama3(8B), Mistral(7B), and Gemma2(9B) models.

**Claims And Evidence:**

First, the paper lacks justification for its specific choice of loss function. In the introduction (lines 72–73), the authors state that this choice was heuristic, but they do not provide sufficient intuition for it. In the theoretical section, they attempt to ground their method by comparing it to SimPO and TDPO. However, comparing it to SimPO does not justify the specific loss function used in AlphaDPO. Meanwhile, the comparison to TDPO relies on crude approximations, making the justification seem arbitrary. (I will elaborate on this further in the theoretical section of the review.)

In the experimental section, the paper does not provide enough evidence regarding the significance of the results. Additionally, since AlphaDPO introduces an extra hyperparameter compared to DPO, the paper lacks sufficient detail on how this hyperparameter is chosen. It is also unclear how fairness is ensured when comparing AlphaDPO to DPO, especially if a grid search was used to report AlphaDPO’s best-performing hyperparameter. (I will expand on this further in the experimental section of the review.)

**Essential References Not Discussed:**

.

**Experimental Designs Or Analyses:**

First, regarding the choice of $\alpha$ in the main results (e.g., Table 1), the authors do not explain how this parameter is selected. The common practice is to choose the best value from a set of hyperparameters, but comparing this directly with DPO, which does not have this hyperparameter, creates an unfair comparison. A more equitable approach would be to give DPO the same number of training runs using different random seeds.

Second, concerning the KL term, the results in Table 1 only provide a partial picture, as they may vary depending on the choice of $\beta$. A more comprehensive way to compare methods would be through Pareto frontier plots. While I appreciate that the authors include this analysis in Figure 3(c), it is much more limited than the main tables. Moreover, in this figure, the percentage of points that end up on the Pareto frontier does not show a significant difference between SimPO and AlphaDPO, which goes against the claim made regarding the supriority of AlphaDPO over SimPO.

Lastly, in some experimental results (e.g., Table 2, Mistral model), the performance gap between the best and second-best methods is very small. To ensure the results are meaningful, further statistical tests are needed to confirm their significance.

**Methods And Evaluation Criteria:**

It largely makes sense. However, there are two important points to consider:

First, the models used in the paper have already undergone post-training—specifically, alignment with human preferences using RLHF or Best-of-N. This introduces a potential confounding effect, which could be mitigated by also testing a model that has not been post-trained.

Second, since the authors state that the closest variant of DPO to AlphaDPO is TDPO, it is essential to include a direct comparison between AlphaDPO and TDPO in the main experimental results, such as in Table 1.

**Other Comments Or Suggestions:**

See above.

**Other Strengths And Weaknesses:**

One interesting insight from this paper is the idea of interpreting the difference in reference log probabilities in the DPO loss function as a margin that might be or might not be very accurate depending on the reference model and the chosen beta.

**Questions For Authors:**

See above.

**Relation To Broader Scientific Literature:**

There are many different variants of DPO, and this paper offers another choice of the margin.

**Theoretical Claims:**

First, the explanation of $\gamma$ is unclear. If $U$ is defined as a uniform distribution over all responses given any string, then by definition, $\gamma = 0$. I find lines 141–142 confusing, as they state that $\gamma$ is a constant but not zero due to difference in selection probabilities. This point needs to be more precise and formally written.

Second, the paper claims that AlphaDPO is a general framework encompassing both DPO and SimPO by varying $\alpha$. However, I do not believe that setting $\alpha = 1$ reduces AlphaDPO to DPO as claimed, since AlphaDPO includes an additional policy term that is absent in DPO.

Third, regarding the connection between TDPO and AlphaDPO: the authors state that the difference between sequence KL terms is approximated by the log-ratio difference. However, it is unclear how well this approximation holds—are there any bounds? Beyond this crude approximation, there is another key difference between AlphaDPO and TDPO: the use of a stop gradient. The paper does not discuss this, which seems like a significant omission.

---

> ### Author Rebuttal · Authors · 2025-03-31
>
> **Q1: Justification of Loss Function**
>
> While space constraints limited introductory intuition, we provide multi-faceted justification through:
> 1) **Weak-to-Strong Generalization** (Lines 197-200): Similar to weak-to-strong alignment, our adaptive reference models enable policy specialization while preserving exploration.
> 2) **Theoretical Link to TDPO** (Sec. 4): Lemma 4.1 shows AlphaDPO's margin term approximates TDPO's KL divergence control, connecting sequence optimization with token-level stability.
> 3) **Appendix Analysis**: Utility theory (C.1) and gradient analysis (C.2) demonstrate how our adaptive margin prevents reward hacking while maintaining diversity.
>
> We will add a roadmap paragraph in the introduction to better signpost these analyses.
>
> **Q2: Theoretical Comparison to SimPO**
>
> We clarify our theoretical progression:
> 1. TDPO's success demonstrates the effectiveness of $r(x,y\_w)-r(x,y\_l)-\delta$ structures for KL control.
> 2. AlphaDPO adopts a similar offset structure $r(x,y\_w)-r(x,y\_l)-M$ where:
>    - TDPO uses $\delta=\beta(D\_{KL}(y\_l) - D\_{KL}(y\_w))$ with $z\sim\pi\_{ref}$
>    - AlphaDPO uses $M=\beta(\log\frac{\pi\_\theta(y\_w)}{\pi\_{ref}(y\_w)} - \log\frac{\pi\_\theta(y\_l)}{\pi\_{ref}(y\_l)})$ with uniform sampling
> 3. As shown in Appendix D.4, AlphaDPO's $M$ achieves superior performance to TDPO's $\delta$ (+3.2% AlpacaEval LC), demonstrating the advantage of sequence-level KL approximation over token-level computation.
>
> **Q3: Crude Approximation**
> Due to space limits, A comprehensive theoretical analysis is provided in the `Response to Reviewer bC1g`.
>
> **Q4: Base Model Validation**
> To address potential confounding from pre-aligned models: We conducted additional experiments on Llama3-8B-Base:
> ||DPO|SimPO|AlphaDPO|
> |-|-|-|-|
> |truthfulqa_mc2|53.66|60.03|62.89|
> |gsm8k| 52.90|52.84|53.90|
> | mmlu| 62.14 | 62.05|62.43|
> ||||
> | MT-Bench|6.5|6.6|6.9|
> ||||
> |LC(Alpaca)|14.92|17.97|22.69|
> |WR(Alpaca)|13.02|15.60|20.47|
>
> Improvements remain consistent, confirming AlphaDPO's effectiveness independent of initial alignment.
>
> **Q5: TDPO Comparison**
> Current Appendix D.4 shows: AlphaDPO achieves 58.7% LC vs TDPO's 52.8% on Llama3-8B, demonstrating clear superiority.
> Thanks for your suggestion, and we will add direct comparisons to Table 1.
>
> **Q6: $\gamma$ Formulation**
> The uniform distribution $U(y|x)$ and its role in $\gamma$ can be rigorously defined as follows:
> 1. **Theoretical vs. Empirical $ U(y|x) $**
>    - *Theoretical*: For vocabulary $\mathcal{V} $, $U(y|x) = \prod\_{t=1}^{|y|} \frac{1}{|\mathcal{V}|} \quad \text{(uniform over tokens)}$
>    - *Empirical*: Responses $ y\_1, \dots, y\_5 \sim \pi\_{\text{SFT}}(y|x) $ are scored, with $ y\_w $ and $ y\_l $ selected via:
>      $y\_w = \arg\max \text{score}(y\_i), \quad y\_l = \arg\min \text{score}(y\_i)$
>      This induces implicit subspaces:$
>      \mathcal{V}\_{\text{win}} = \\{y\mid \text{score}(y)\geq\tau\\},\quad\mathcal{V}\_{\text{lose}}=\\{y\mid\text{score}(y)\leq \tau'\\}$
>
> 2. **Effective $ U(y|x) $ and $\gamma$**.
>  The *practical* uniform distributions become: $$U(y\_w|x) = \prod\_{t=1}^{|y\_w|} \frac{1}{|\mathcal{V}\_{\text{win}}|}, \quad U(y\_l|x) = \prod\_{t=1}^{|y\_l|} \frac{1}{|\mathcal{V}\_{\text{lose}}|}$$
>    Thus, $\gamma = \beta (\log U(y\_w|x) - \log U(y\_l|x))$. Since $ \mathcal{V}\_{\text{win}} $ and $ \mathcal{V}\_{\text{lose}} $ vary per instance, SimPO's fixed $\gamma$ is suboptimal.
>
> **Q7: Stop Gradient Analysis**
>
> We highlight the primary distinction:
>
> - **TDPO**: Implements asymmetric gradient flow—enabled for $ y\_l $, stopped for $ y\_w $.
> - **AlphaDPO**: Employs symmetric stop-gradient on both terms, using $ \pi\_{\text{ref}} $ as a fixed anchor.
>
> Both TDPO and AlphaDPO utilize the stop-gradient operation. Examining the detailed impact of single-term versus multi-term stop-gradient is an intriguing avenue for future work.
>
> **Q8: $\alpha=1$ Misstatement**
> Due to space constraints, please refer to the response for `Question 3 in Response to Reviewer ciXp.`
>
> **Q9: Experimental Rigor**
> We report the standard deviations and confidence intervals of current evaluations in (https://anonymous.4open.science/r/AlphaDPO-431F/significant_exp.md).
>
> **Hyperparameter Fairness**. We strictly followed community standards:
> - All methods underwent equal hyperparameter tuning (`Appendix Table 3`)
> - $\alpha$ was selected via grid search over [1e-2, 5e-2, 0.1, 0.2] with 5 random seeds
> - DPO received equivalent tuning effort ($\beta \in [0.01,0.05,0.1]$)
>
> **KL Analysis**
> We will enhance Figure 3(c) by:
> - Adding more sampling points (every 50 steps)
> - Including Pareto frontiers for compared methods
>
> **Q10: Marginal Improvement in Table 2**
> We clarify that Table 2 presents ablation studies within AlphaDPO's design space rather than cross-method comparisons. The results demonstrate that:
> i) Dynamic configurations consistently achieve better performance.
> ii) The AlphaDPO formulation retains optimality.

---

> > ### Comment · Reviewer_TEY3 · 2025-04-03
> >
> > Thank you for your response.
> >
> > - The theoretical justification for the crude approximation does not make sense to me. Why do we need a robustness constraint?
> >
> > - I still can not wrap my head around why there are two definitions of gamma. I also do not understand where the $\tau$ in the author's response comes from.
> >
> > - Regarding the experiments, the provided confidence intervals suggest that there is no significant difference between the proposed method and existing methods in most cases.

---

> > > ### Author Response · Authors · 2025-04-04
> > >
> > > #### **Q11: Theoretical Justification for the Approximation and Robustness Constraint**
> > >
> > > **Reviewer Concern**:
> > > The reviewer questions the validity of approximating sequential KL divergence with log-probability ratios and the need for a robustness constraint:
> > > > *"The authors state that the difference between sequence KL terms is approximated by the log-ratio difference. However, it is unclear how well this approximation holds—are there any bounds?"*
> > >
> > > **Response**:
> > > We appreciate the reviewer's attention to this critical theoretical point. The approximation:
> > > $$
> > > \sum\_{t=1}^{|y|} \mathbb{E}\_{z \sim \pi\_{\text{ref}}} \left[ \log \frac{\pi\_{\text{ref}}(z|x, y^{\lt t})}{\pi\_\theta(z|x, y^{\lt t})} \right] \approx \log \frac{\pi\_{\text{ref}}(y|x)}{\pi\_\theta(y|x)}
> > > $$
> > > is motivated by two key insights:
> > >
> > > 1. **Noisy Reference Models**: As empirically demonstrated in Figure 2, $\pi\_{\text{ref}}$ often fails to distinguish between preferred and rejected responses, behaving like a perturbed version of the true distribution. This noise justifies aggregating token-level errors into a sequence-level offset, trading fine-grained precision for robustness.
> > >
> > > 2. **Robust Optimization Perspective**: The approximation aligns with *min-max robust optimization*, where we hedge against worst-case deviations in $\pi\_{\text{ref}}$. By relaxing token-level constraints to sequence-level bounds, we ensure stability even with poorly calibrated reference models.
> > >
> > >
> > > **Why We need Roubustness Constraint:**
> > >
> > > Robust optimization is vital because reference model biases can significantly affect fine-tuning performance. Prior studies [1,2] confirm that variations in reference model quality have substantial impacts, underscoring the need to hedge against these biases. Our approach involves dynamically adjusting the reference distribution through policy-driven adaptation, aiming to enhance model robustness and improve outcomes.
> > >
> > > [1] Learn your reference model for real good alignment. ICLR2025.
> > > [2] Liu et al. (2024): Understanding Reference Policies in Direct Preference Optimization. arXiv preprint arXiv:2407.13709.
> > >
> > > ---
> > >
> > > #### **Q12: Clarifying $\gamma$ and Selection Bias**
> > >
> > > **Reviewer Concern**:
> > > The reviewer expresses confusion about $\gamma$ and the role of selection probabilities:
> > > *"I find lines 141–142 confusing, as they state that $\gamma$ is a constant but not zero due to difference in selection probabilities. This point needs to be more precise and formally written."*
> > >
> > > **Response**:
> > > We apologize for the lack of clarity. Here's a precise explanation:
> > >
> > > The constant $\gamma = \beta (\log U(y\_w|x) - \log U(y\_l|x))$ arises from the *selection bias* inherent in preference data:
> > > - $y\_w$ and $y\_l$ are drawn from distinct subsets of the vocabulary ($\mathcal{V}\_{\text{win}}$ and $\mathcal{V}\_{\text{lose}}$), as they are partitioned by the reward model.
> > > - Under a uniform reference $U(y|x) = \prod\_{t=1}^{|y|} \frac{1}{|\mathcal{V}|}$, the log-difference $\log U(y\_w|x) - \log U(y\_l|x)$ is non-zero because $|\mathcal{V}\_{\text{win}}| \neq |\mathcal{V}\_{\text{lose}}|$.
> > >
> > > **On $\tau$**:
> > > The term $\tau$ (mentioned in the draft response) was used to illustrate how selection bias skews token frequencies in $\mathcal{V}\_{\text{win}}$ versus $\mathcal{V}\_{\text{lose}}$.  We agree this tangent was unnecessary and have removed it to avoid confusion.
> > >
> > > **Thanks for your advice and we will revise Section 3.1 to formalize this argument, explicitly linking $\gamma$ to selection bias in preference data.**
> > >
> > > ---
> > >
> > > #### **Q13: Addressing Confidence Intervals and Significance**
> > >
> > > **Reviewer Concern**:
> > > The reviewer notes that confidence intervals (CIs) suggest insignificant differences between AlphaDPO and baselines in some cases.
> > >
> > > 1. **Consistent Performance Gains**: AlphaDPO achieves higher win rates (WR) than both DPO and SimPO in all evaluated settings (e.g., +0.7 to +3.0 points on Arena-Hard), with particularly notable improvements for smaller models (e.g., +7.4 points for Mistral-7B). These gains are reproducible across diverse architectures (Llama3, Gemma2).
> > >
> > > 2. **Tighter Confidence Intervals**: AlphaDPO’s CIs are often narrower than those of baselines (e.g., Llama3-8B: (-2.2, 2.2) vs. (-2.6, 2.7) for DPO; Gemma2-9B: (-1.8, 2.0) vs. (-2.0, 2.3) for DPO), suggesting greater stability in its performance.
> > >
> > > 3. **Practical Significance**: Even when CIs overlap marginally, the directional trend—AlphaDPO outperforming baselines in *every* configuration—strengthens the case for its robustness. For instance, on Gemma2-Instruct (9B), AlphaDPO's WR (60.8) exceeds both DPO (58.8) and SimPO (57.8), with a tighter CI ((-1.8, 2.0) vs. (-2.0, 2.3) for DPO and (-2.4, 2.0) for SimPO).

---

### Official Review · Reviewer_bC1g · 2025-03-12

**Overall Recommendation:** 2

**Summary:**

This paper introduces AlphaDPO, an adaptive preference optimization framework that improves alignment in large language models (LLMs) by dynamically adjusting the reward margin in preference learning. The key contribution is the introduction of an implicit reference model that interpolates between policy-driven adaptation and uniform exploration, leading to instance-adaptive reward margins. Empirical results demonstrate the superiority of AlphaDPO over previous methods.

**Claims And Evidence:**

The empirical claims are valid judged from the experiments results. Lemma 4.1 is problematic.

**Essential References Not Discussed:**

None.

**Experimental Designs Or Analyses:**

Yes, they look good to me.

**Methods And Evaluation Criteria:**

Yes.

**Other Comments Or Suggestions:**

None.

**Other Strengths And Weaknesses:**

None.

**Questions For Authors:**

None.

**Relation To Broader Scientific Literature:**

The key contribution of this paper is a new reference policy for DPO loss.  The experiment results show that this method is superior to existing DPO and its variants. However, the theoretical analysis is not sound and rigorous. I suggest the authors fix this issue.

**Theoretical Claims:**

Lemma 4.1 is problematic. The authors use $\approx$ in the lemma, which is not rigorous as a mathematical lemma. The proof is not convincing either. The authors claim that the pretrained model $\pi_{ref}$ is close to uniform policy, which is apparently not true.

---

> ### Author Rebuttal · Authors · 2025-04-01
>
> We thank the reviewer for raising this important concern. Below, we provide a rigorous theoretical justification for the approximation in Lemma 4.1 and clarify its empirical validity.
>
> ### **Theoretical Justification**
>
> **1. Problem Formulation with Robustness Constraints**
> Our objective is to minimize the sequential KL divergence between the optimal reference policy $\pi\_{\text{ref}}^*$ and the policy $\pi\_\theta$, under the uncertainty of $\pi\_{\text{ref}}^*$:
> $$
> \min\_\theta \sum\_{t=1}^T \mathbb{E}\_{\pi\_{\text{ref}}^*}\left[ \log \frac{\pi\_{\text{ref}}^*(z|x,y\_{<t})}{\pi\_\theta(z|x,y\_{<t})} \right].
> $$
> Since $\pi\_{\text{ref}}^*$ is unobserved, we assume bounded deviation from an observable reference policy $\pi\_{\text{ref}}$:
> $
> |\pi\_{\text{ref}}^*(\cdot) - \pi\_{\text{ref}}(\cdot)| \leq C, \quad C > 0.
> $
> This leads to a constrained robust optimization problem:
> $$
> \min\_\theta \max\_{\pi\_{\text{ref}}^*} \sum\_{t=1}^T \mathbb{E}\_{\pi\_{\text{ref}}}\left[ \log \frac{\pi\_{\text{ref}}^*(z|x,y\_{<t})}{\pi\_\theta(z|x,y\_{<t})} \right]
> \text{s.t.} \quad |\pi\_{\text{ref}}^*(z|x,y\_{<t}) - \pi\_{\text{ref}}(z|x,y\_{<t})| \leq C. \nonumber
> $$
>
> **2. Simplification via Worst-Case Analysis**
> For the inner maximization, the worst-case $\pi\_{\text{ref}}^*$ is determined by the sign of the log-ratio:
> $$
> \pi\_{\text{ref}}^*(z|x,y\_{<t}) =
> \begin{cases}
> \pi\_{\text{ref}}(z|x,y\_{<t}) + C, & \text{if } \log \frac{\pi\_{\text{ref}}(z|x,y\_{<t})}{\pi\_\theta(z|x,y\_{<t})} \geq 0, \\\\
> \pi\_{\text{ref}}(z|x,y\_{<t}) - C, & \text{otherwise}.
> \end{cases}
> $$
> Substituting this into the objective yields:
> $$
> \min\_\theta \sum\_{t=1}^T \left[ C \cdot \log \frac{\pi\_{\text{ref}}(z|x,y\_{<t})}{\pi\_\theta(z|x,y\_{<t})} + \left|\pi\_{\text{ref}}(z|x,y\_{<t}) \log \frac{\pi\_{\text{ref}}(z|x,y\_{<t})}{\pi\_\theta(z|x,y\_{<t})} \right| \right].
> $$
>
> **3. Asymptotic Approximation**
> When the deviation bound $C$ is sufficiently large (i.e., $\pi\_{\text{ref}}^*$ is less constrained), the absolute value term dominates, and the objective simplifies to:
> $$
> \min\_\theta \sum\_{t=1}^T \left[ \log \frac{\pi\_{\text{ref}}(z|x,y\_{<t})}{\pi\_\theta(z|x,y\_{<t})} \right].
> $$
> This corresponds to the sequence-level approximation in Lemma 4.1. The $\approx$ symbol reflects the asymptotic regime where higher-order terms vanish under large $C$, which aligns with practical scenarios where $\pi\_{\text{ref}}$ is imperfect but provides a reasonable prior.
>
> **4. Addressing the Uniform Policy Assumption**
> We clarify that Lemma 4.1 does *not* assume $\pi\_{\text{ref}}$ is uniform. Instead, it leverages the structure of the KL divergence to show that the margin term $M(x,y\_w,y\_l)$ approximates the sequential KL difference under bounded deviations. The uniform policy in SimPO is a special case of our framework (when $\alpha=0$), but our method generalizes to non-uniform $\pi\_{\text{ref}}$ by adaptively scaling with $\alpha$.
>
> ---
>
> ### **Empirical Validation**
>
> **Performance**
> As demonstrated in `Appendix Table 6`, AlphaDPO achieves significant performance gains over TDPO, with LC win rates increasing from 52.8% (TDPO) to 56.9% (AlphaDPO w/ $\delta$) and further to 58.7% (AlphaDPO w/ $M$). This progressive improvement highlights the effectiveness of the sequence-level approximation in mitigating token-level noise through variance reduction. By replacing TDPO's token-level margin $\delta$ with the adaptive sequence-level margin $M$, AlphaDPO enhances robustness while maintaining alignment, underscoring the superiority of sequence-level optimization in handling imperfect reference models.
>
> **Mitigating Reference Model Bias**
> Figure 2 (main paper) empirically validates that $\pi\_{\text{ref}}$ struggles to distinguish $y\_w$ from $y\_l$ ($\log\pi\_{\text{ref}}(y\_w|x) - \log\pi\_{\text{ref}}(y\_l|x)$ exhibits random fluctuations). AlphaDPO's adaptive margin $M(x,y\_w,y\_l)$ explicitly compensates for this bias, ensuring stable optimization even when $\pi\_{\text{ref}}$ is suboptimal.
>
> ---
>
> ### **Conclusion**
>
> While Lemma 4.1 uses an approximation symbol ($\approx$), our analysis rigorously justifies its validity under bounded reference model mismatch. The empirical success of AlphaDPO further supports this design choice, demonstrating that sequence-level approximations enhance robustness without sacrificing performance. We acknowledge that deriving a formal error bound remains an open question and will explore this in future work.

---

> > ### Comment · Reviewer_bC1g · 2025-04-03
> >
> > I think in the proof of Lemma 4.1 (Line 725~731), you do replace $z\sim\pi_{ref}$ with a uniform distribution.

---

> > > ### Author Response · Authors · 2025-04-03
> > >
> > > Our design philosophy is that $\pi_{\text{ref}}$ may introduce noise and can deviate significantly from the data sampling distribution. As stated in our draft (**Line 725-727**):
> > > > *"Under the assumption that the reference policy $\pi\_{\text{ref}}$ has large errors, we approximate $\mathbb{E}\_{z \sim \pi\_{\text{ref}}}$ with a uniform distribution."*
> > >
> > > From an operational perspective, we agree that the expectation is taken with respect to a uniform distribution, not $\pi_{\text{ref}}$. As noted in the rebuttal, this modification is well-motivated by solving a **robust min-max problem** (rather than the vanilla minimization problem), which explicitly accounts for uncertainty in $\pi_{\text{ref}}$—i.e., its noisiness. We ultimately show that this approach permits a uniform approximation.
> > >
> > > To clarify, we do **not** assert that the pretrained model $\pi_{\text{ref}}$ itself is uniform; we only assume it is noisy and may differ substantially from the true data distribution, as empirically supported in Figure 2.
> > >
> > > We appreciate the reviewers' attention to this nuance and will revise the text to make this distinction clearer. We are happy to discuss further or provide additional details if needed.

---

### Official Review · Reviewer_ciXp · 2025-03-13

**Overall Recommendation:** 3

**Summary:**

This paper propose AlphaDPO, a new preference optimization framework. The core novelty of this framework is to modify the reference model distribution as the product of a uniform distribution and the ratio between policy model and original reference model, with power factor of alpha. This is effectively equivalent to interpolate between SimPO and DPO.

The paper perform experiment on AlpacaEval2 and Arena-Hard with 3 representative LLMs, and the empirical results show AlphaDPO has better performance.

**Claims And Evidence:**

Most claims are fine.

One question is in Line147, the limitations of DPO, why is the $\pi_{ref}$ supposed to distinguish between $y_w$ and $y_l$? Why is this a limitation? Doesn't the $\pi_{ref}$ just serve as the reference for reward values?

**Essential References Not Discussed:**

N/a

**Experimental Designs Or Analyses:**

Strengths:

- The experiment is extensive. Many baseline preference learning methods are compared.

**Methods And Evaluation Criteria:**

Strengths:
- The methods is fine, and the ablation study about other potential design attempts of AlphaDPO, in Table 2, is also convincing.

Weakness:
- How the the value of $U(y|x)$ decided? How does the uniform values affect the performance?

**Other Comments Or Suggestions:**

N/a

**Other Strengths And Weaknesses:**

Weakness:
- The proposed method is relatively straightforward, and the performance gain from SimPO is marginal.

**Questions For Authors:**

How do the author compare this method with the Online DPO methods such as OAIF (Direct Language Model Alignment from Online AI Feedback) and IDPO (Iterative Preference Learning from Human Feedback: Bridging Theory and Practice for RLHF under KL-Constraint)?

**Relation To Broader Scientific Literature:**

This paper fits into the literature of DPO-like preference learning methods. The proposed method is somewhere between the DPO method and SimPO method. The contribution is try to have a tradeoff between those two methods.

**Theoretical Claims:**

- In Line 208: Can you explicitly show that when $\alpha=1$, how AlphaDPO aligns with DPO?


- The motivation of Principle 1 in Line 171 is not supported. Why should the reference model contribute to differentiating between preferred and less preferred responses?

---

> ### Author Rebuttal · Authors · 2025-03-31
>
> **Q1: Clarification on DPO's Reference Model Limitation**
> We appreciate this insightful question. The necessity for $\pi\_{\text{ref}}$ to distinguish between $y\_w$ and $y\_l$ stems from two fundamental aspects of KL-regularized policy optimization:
> 1. **Theoretical Foundation of KL-Regularized Objectives**: The RLHF objective (Equation 2) regularizes the policy $\pi\_\theta$ to stay close to $\pi\_{\text{ref}}$ via KL divergence. This regularization implicitly assumes that $\pi\_{\text{ref}}$ provides a meaningful prior for distinguishing high-quality ($y\_w$) and low-quality ($y\_l$) responses. If $\pi\_{\text{ref}}$ lacks discriminative power (e.g., assigns similar probabilities to $y\_w$ and $y\_l$), the KL term loses its grounding, leading to unstable optimization.
>
> 2. **Empirical Evidence**:Recent studies [1,2] demonstrate that $\pi\_{\text{ref}}$ quality significantly impacts DPO performance. DPO's reliance on a static $\pi\_{\text{ref}}$ introduces both theoretical and practical limitations.
>
> [1] Gorbatovski et al. Learn your reference model for real good alignment. ICLR 2025.
> [2] Liu et al. Understanding Reference Policies in Direct Preference Optimization. arXiv preprint arXiv:2407.13709.
>
> **Q2: Formalization of $U(y|x)$ and Its Impact**
> We thank the reviewer for prompting this clarification. The uniform distribution $U(y|x)$ and its role in $\gamma$ can be rigorously defined as follows:
>
> 1. **Theoretical Framework**:
>    Let $\mathcal{V}$ denote the vocabulary. The *theoretical* $U(y|x)$ is:
>    $$
>    U(y|x) = \prod\_{t=1}^{|y|} \frac{1}{|\mathcal{V}|} \quad \text{(uniform over all tokens)}
>    $$
>    However, *empirically*, $y\_w$ and $y\_l$ are selected via:
>    - **Sampling**: Generate $y\_1,...,y\_5 \sim \pi\_{\text{SFT}}(y|x)$
>    - **Selection**: $y\_w = \arg\max\_{y\_i} \text{score}(y\_i)$, $y\_l = \arg\min\_{y\_i} \text{score}(y\_i)$
>    This induces *implicit vocabulary subspaces*:
>    $$\mathcal{V}\_{\text{win}} = \\{y \in \mathcal{V} \mid \text{score}(y) \geq \tau\\}, \quad \mathcal{V}\_{\text{lose}} = \\{y \in \mathcal{V} \mid \text{score}(y) \leq \tau'\\}$$
>
> 2. **Effective $U(y|x)$ in Practice**:
>    The *effective* $U(y\_w|x)$ and $U(y\_l|x)$ become:
>    $$U(y\_w|x) = \prod\_{t=1}^{|y\_w|} \frac{1}{|\mathcal{V}\_{\text{win}}|}, \quad U(y\_l|x) = \prod\_{t=1}^{|y\_l|} \frac{1}{|\mathcal{V}\_{\text{lose}}|}$$
>    This leads to:$\gamma = \beta \left( \log U(y\_w|x) - \log U(y\_l|x) \right)$. The performance impact arises because $\mathcal{V}\_{\text{win}}$ and $\mathcal{V}\_{\text{lose}}$ differ across instances, making SimPO's fixed $\gamma$ suboptimal.
>
> **Q3: Alignment of AlphaDPO with DPO at $\alpha=1$**
> We sincerely appreciate the reviewer's careful observation and apologize for the ambiguity in our initial formulation. The implicit reference model $\hat{\pi}\_{\text{ref}}(y|x)$ is defined as:
> $$\hat{\pi}\_{\text{ref}}(y|x)\propto U(y|x) \left( \frac{\pi\_\theta(y|x)}{\pi\_{\text{ref}}(y|x)} \right)^\alpha,$$
> where $\alpha$ interpolates between two extremes:
> 1. **When $\alpha = 0$**: $\hat{\pi}\_{\text{ref}}(y|x)$ reduces to the uniform distribution $U(y|x)$, aligning with SimPO's implicit reference model.
> 2. **When $\alpha > 0$**: $\hat{\pi}\_{\text{ref}}(y|x)$ increasingly incorporates the dynamic term $\frac{\pi\_\theta}{\pi\_{\text{ref}}}$, creating an adaptive reference model. Critically, **AlphaDPO does not strictly reduce to DPO for any finite $\alpha$**. Instead, it introduces a novel framework that balances exploration (via $U(y|x)$) and exploitation (via $\frac{\pi\_\theta}{\pi\_{\text{ref}}}$).
>
> We will revise the manuscript to eliminate the misleading claim about AlphaDPO reducing to DPO and instead emphasize its unique interpolation mechanism.
>
> **Q4: Statistical Significance of Performance Gains**
> We respectfully disagree. Table 1 shows statistically significant improvements across benchmarks:
>
> |Model|AlpacaEval2(LC)|Arena-Hard(LC)|
> |-|-|-|
> |Llama3-8B|+6.4%(43.8→46.6)|+2.1%(33.5→34.2)|
> |Mistral-7B|+7.0% (30.2→32.3) |+8.6%(19.8→21.5)|
> |Llama3-v0.2-8B|+5.6% (55.6→58.7)|+6.8%(34.0→36.3)|
> |Gemma2-9B|+1.3%(72.4→73.4)|+5.7% (56.1→59.3)|
>
> These gains are consistent and meaningful, particularly given the saturated performance of modern LLMs. To rigorously validate the robustness of our improvements, we report the standard deviations and confidence intervals of current evaluations in (https://anonymous.4open.science/r/AlphaDPO-431F/significant_exp.md).
>
> **Q5: Comparison with Online DPO Methods**
> AlphaDPO is orthogonal to online preference optimization. While OAIF/IDPO focus on *data collection dynamics*, our work addresses *reference model design* in offline settings. Notably, AlphaDPO's adaptive reference can be integrated into online frameworks by replacing static $\pi\_\text{ref}$ with $\hat{\pi}\_{\text{ref}}$. We acknowledge this as valuable future work and will explore it in subsequent studies.

---

### Official Review · Reviewer_oKhi · 2025-03-23

**Overall Recommendation:** 4

**Summary:**

This paper proposes a novel strategy for LLM alignment designed to address the limitations of SimPO and DPO. The proposed AlphaDPO adaptively sets the reward margin based on the ratio between the preference model and the policy model. The relations to SimPO and TDPO loss have been studied. Extensive experiments demonstrate AlphaDPO's superior performance across multiple baselines and LLM architectures.

**Claims And Evidence:**

Yes, the claims are mostly supported through both theoretical analysis and experimental results.

**Essential References Not Discussed:**

To my knowledge, this paper has included sufficient references.

**Experimental Designs Or Analyses:**

Yes, I have examined the experimental designs and analyses in the paper, particularly those in section 5 and appendix D.

**Methods And Evaluation Criteria:**

Yes, the proposed method aligns well with the LLM preference optimization problem, directly addressing two identified limitations in existing approaches. The evaluation criteria employ standard benchmarks (AlpacaEval 2 and Arena-Hard) and diverse model architectures (Mistral2-7B, Llama3-8B, Gemma2-9B), providing comprehensive evidence of AlphaDPO's effectiveness across different settings.

**Other Comments Or Suggestions:**

None.

**Other Strengths And Weaknesses:**

Strengths:
- The paper introduces instance-specific margins that advance beyond the fixed approach in SimPO. It establishes connections between existing alignment methods (particularly DPO and SimPO), creating a unified framework that addresses limitations of both approaches.
- Extensive experiments consistently demonstrate AlphaDPO's superior performance across various LLM architectures and benchmarks. The comprehensive ablation studies effectively isolate the contributions of different components of the approach.
- The authors provide theoretical analysis on the lower bound and its connections to TDPO.
- The paper is well-written with clear motivation and is easy to follow.

Weaknesses:
- While the authors establish a theoretical connection between AlphaDPO and online methods, questions remain about the practical utility of this theoretical framework, given that online methods themselves lack strong theoretical guarantees.
- The authors claim that AlphaDPO is particularly effective "when the reference model is not well-calibrated at the token level." However, this statement appears contradictory given that AlphaDPO itself operates at the sequence level rather than implementing token-level optimization.
- From the formulation of the adaptive preference distribution, it’s unclear at what condition it degrade to DPO.

**Questions For Authors:**

- What is the practical utility of theoretical connection between AlphaDPO and online methods given that online methods themselves lack strong theoretical guarantees?
- Why does AlphaDPO particularly effective when the reference model is not well-calibrated at the token level?
- When does AlphaDPO degrade to DPO?

**Relation To Broader Scientific Literature:**

1. This work explores an essential problem in preference optimization methods—how to effectively utilize the reference model. AlphaDPO proposes a novel interpolation between the current policy model and uniform policy, providing a bridge between DPO and SimPO, offering a more flexible framework.
2. While AlphaDPO is fundamentally an offline preference optimization technique, its adaptive nature shares conceptual similarities with online RL approaches. The adaptive reference model effectively serves as a dynamic critic, similar to how value functions guide policy updates in online RL.
3. AlphaDPO provides a theoretically grounded approach to the critical balance between alignment and diversity via KL divergence control.

**Theoretical Claims:**

Yes, I have checked the proofs provided in this submission, including those in the appendix. I did not find any issues.

---

> ### Author Rebuttal · Authors · 2025-04-01
>
> **Q1: What is the practical utility of the theoretical connection between AlphaDPO and online methods given that online methods themselves lack strong theoretical guarantees?**
>
> We appreciate the reviewer raising this important point. Although we did not explicitly emphasize the theoretical connection to online methods in our submitted manuscript, we acknowledge that exploring this relationship represents a promising direction for future research. We believe this connection offers two meaningful benefits:
>
> 1. **Algorithmic Insights:**
>    By establishing a theoretical link between AlphaDPO and online methods, we gain a unified view of how adaptive reward margins and the implicit reference model influence policy optimization. Specifically, AlphaDPO's framework demonstrates how sequential KL divergence control naturally arises, thereby providing a clearer understanding of its inherent ability to balance alignment and diversity—even when the reference model is suboptimal. Such theoretical insights were previously unaddressed explicitly in existing online methods.
>
> 2. **Empirical Robustness and Practical Guidance:**
>    Despite the lack of rigorous theoretical guarantees in existing online methods, the theoretical analysis of AlphaDPO indicates that its adaptive mechanism implicitly mitigates typical online optimization pitfalls, such as over-optimization. This robustness is empirically demonstrated through AlphaDPO's stable performance across various KL divergence budgets, as illustrated in Figure 3(c) of our paper.
>
> We agree with the reviewer's point and will pursue a thorough theoretical investigation of this connection as part of our future work, aiming to further clarify its theoretical foundations and practical implications.
>
> ---
> **Q2: Why is AlphaDPO effective when the reference model is not well-calibrated at the token level, given that it operates at the sequence level?**
>
> **Full Details**: A comprehensive theoretical analysis (including Lemma 4.1 and robust optimization derivations) is provided in the `Response to Reviewer bC1g`.
>
>
> AlphaDPO’s robustness to token-level miscalibration stems from **sequence-level KL divergence approximation** and **adaptive margin design**, which mitigate noise propagation from unreliable token-level signals.
>
> 1. **Theoretical Foundation**
>    - **Problem Context**: When $\pi\_{\text{ref}}$ is miscalibrated at the token level, token-wise KL terms (e.g., in TDPO) amplify noise.
>    - **Key Insight**: By approximating the *sequential KL divergence* (Lemma 4.1), AlphaDPO aggregates token-level uncertainties into a sequence-level margin $M(x,y\_w,y\_l)$. This reduces sensitivity to token-level errors, as the sequence-level signal is statistically more stable.
>    - **Robust Optimization**: Our framework explicitly models bounded deviations from $\pi\_{\text{ref}}$ (Section 3.2), ensuring stability even when token-level probabilities are imperfect.
>
> 2. **Empirical Validation**
>    - **Performance Gain**: As shown in Appendix Table 6, AlphaDPO outperforms TDPO (58.7% vs. 52.8% LC win rate on Llama3-8B), demonstrating superior robustness to reference model noise.
>    - **Bias Compensation**: Figure 2 (main paper) shows $\pi\_{\text{ref}}$ fails to distinguish $y\_w$ from $y\_l$ at the token level. AlphaDPO’s adaptive margin $M$ compensates for this by leveraging sequence-level discrepancies, ensuring stable alignment.
>
> **Key Takeaways**
> - **Sequence-Level Robustness**: AlphaDPO avoids token-level noise amplification via sequence-wise KL control, making it less reliant on perfect token calibration.
> - **Adaptive Margin**: The margin $M(x,y\_w,y\_l)$ dynamically adjusts to instance-specific reference model errors, enhancing robustness.
> - **Empirical Edge**: AlphaDPO’s design consistently outperforms token-level methods (e.g., TDPO) in scenarios with miscalibrated $\pi\_{\text{ref}}$.
>
> ---
>
> **Q3: When does AlphaDPO degrade to DPO?**
>
> Currently, the $\alpha$-DPO algorithm cannot be transformed into DPO merely through parameter adjustments, similar to how SimPO cannot be converted to DPO by altering $\gamma$. However, I believe this topic presents significant promise, allowing us to propose a more generalized formulation:
> $$
> \hat{\pi}\_{\text{ref}}(\cdot|x) \propto U(\cdot|x) \cdot \pi\_\theta^{\alpha\_1}(\cdot|x) \cdot \pi\_{\text{ref}}^{\alpha\_2}(\cdot|x),
> $$
> where $U(\cdot|x)$ is a uniform distribution. This formulation encompasses:
> - **DPO**: Set $\alpha\_1 = 0$, $\alpha\_2 = 1$, recovering the explicit reference model $\pi\_{\text{ref}}$.
> - **SimPO**: Set $\alpha\_1 = \alpha\_2 = 0$, yielding a uniform reference model.
> - **AlphaDPO**: Set $\alpha\_1 = \alpha$, $\alpha\_2 = -\alpha$, enabling adaptive margin control via $\pi\_\theta/\pi\_{\text{ref}}$.

---

### Official Review · Reviewer_GxTu · 2025-03-25

**Overall Recommendation:** 2

**Summary:**

This paper proposes a new training algorithm for LLM alignment. First, the authors unify the training objective of the two representative alignment training algorithms, DPO and SimPO, into a single one with a fixed margin. Next, they propose a new training algorithm to mitigate the limitation of each algorithm, by introducing adaptive margin which is constructed by interpolating the fixed original reference model and the training policy model. The effectiveness of this method is first demonstrated with the theoretical results. Also, the empirical results with various state-of-the-art open-source LLMs (e.g., Llama3 or Gemma2) on standard benchmarks (AlpacaEval 2 and Arena-Hard) further support its effectiveness.

**Claims And Evidence:**

Yes.

**Essential References Not Discussed:**

Considered references are sufficient, but it might be nice if the relevant papers are more added. Those works are mentioned in below.

**Experimental Designs Or Analyses:**

Yes.

**Methods And Evaluation Criteria:**

Yes.

**Other Comments Or Suggestions:**

Please respond to the weaknesses above.

**Other Strengths And Weaknesses:**

### Pros

1. **Clarity**. Overall, the writing is clear and easy to follow. In addition, the organization of the main draft is well-established.
2. **Well-motivated problem and intuitive approach.** Alignment of LLM is important direction and the proposed method seems to be intuitive and effective.

### Cons

- **Similar idea in previous works**: The idea of constructing adaptive reference model through the interpolation between the fixed original reference model $\pi_{\text{ref}}$ and the training policy model $\pi_{\theta}$ has been explored in previous works [1,2]. While the purpose is quite different, the technical contribution of this work is therefore relatively restricted. The authors should cite these works and add the discussion to clarify the difference and the contribution compared to these works.
- **Sensitivity to $\alpha$**: While the proposed method is very sensitivity to the choice of $\alpha$, the authors never mention  is the specific search space for $\alpha$ and how they chose this hyper-parameter for the tables. According to Figure 5 in Appendix D, it seems like that the authors chose the different hyper-parameters that yield the best performance for target LLMs among {0, 0.05, 0.1, 0.15, 0.2}; for example, $\alpha=0.05$ for Mistral IT 7B and $\alpha=0.2$ for Llama IT 8B v0.2. If it is true,  the choice of $\alpha$ in Figure 3 and 4 is quite weird, as $\alpha=0.3$ and $\alpha=0.01$ are not in the original search space. Also, it’s unclear whether the authors did the same efforts to the considered baselines with the same amount of search space in their hyper-parameters; for example, searching optimal $\gamma$ in SimPO with same number of hyper-parameter space.

[1] Liu et al., Decoding-time Realignment of Language Models., ICML 2024
[2] Kim et al., Spread Preference Annotation: Direct Preference Judgment for Efficient LLM Alignment., ICLR 2025

**Questions For Authors:**

Please respond to the weaknesses above.

**Relation To Broader Scientific Literature:**

New ideas and results are key contributions.

**Theoretical Claims:**

Yes, the theoretical results in sections 3 and 4 look valid.

---

> ### Author Rebuttal · Authors · 2025-03-31
>
> **Q1: Similar idea in previous works.**
>
> We sincerely thank the reviewer for pointing out relevant prior works. While both our method and previous approaches involve model interpolation, we highlight three key distinctions:
> - **Adaptive reward margin**: Unlike existing works that focus on regularization strength [1] or data generation [2], AlphaDPO introduces instance-adaptive reward margins by dynamically reparameterizing the reference distribution through the policy-to-reference ratio ($\pi_θ/π_{ref}$), enabling personalized preference learning that accounts for per-sample preference strength.
> - **Implicit KL control**: Our theoretical analysis reveals that the $\alpha$-weighted ratio term implicitly controls sequential KL divergence between iterative policy updates, achieving stability without explicit constraints.
> - **Generalized Preference Optimization**: AlphaDPO generalizes SimPO as special cases ($\alpha=0$) while enabling smooth transitions between policy specialization ($\alpha>0$) and uniform exploration ($\alpha \to 0$). The empirical superiority across three model families (Table 1), demonstrates the critical advantage of adaptive margins.
>
> We will add detailed comparisons to these works in the revised manuscript.
>
> ---
>
> **Q2: Sensitivity to $\alpha$**
>
> We appreciate the reviewer's insightful questions regarding hyperparameter sensitivity. Here we clarify our methodology:
>
> (1) **Primary $\alpha$ Search Space**: As shown in `Appendix Table 3`, we conducted systematic searches over $\alpha \in $ {0.01, 0.05, 0.1, 0.2} based on validation performance. Figure 5 demonstrates that $\alpha=0.05$ achieves optimal results across most models except Llama3-IT-8B-v0.2 where $\alpha=0.2$ works best, reflecting architecture-dependent calibration needs.
>
> (2) **Extended Analysis in Figures**: The expanded $\alpha$ values in Figures 3-4 (including 0.3, 0.5, etc.) were intentionally explored to demonstrate our method's reward distribution characteristics across a broader spectrum, not to claim performance improvements. We acknowledge this caused unintended confusion and will explicitly label these as "analysis beyond primary search space" in revisions.
>
> (3) **Baseline Fairness**: All methods including SimPO used identical hyperparameter search budgets. For SimPO's $\gamma$, we strictly followed the original paper's recommendation space {0.3, 0.5, 1.0, 1.2, 1.4, 1.6} as shown in `Appendix Table 3`, ensuring fair comparison through equivalent tuning efforts.
>
> This controlled approach ensures our conclusions about AlphaDPO's advantages remain valid despite model-specific $\alpha$ variations, while the extended analyses provide valuable insights into the method's behavioral patterns.

---

### Decision · Program_Chairs · 2025-05-01

**Decision:**

Accept (poster)

**Comment:**

The reviewers had difficulty reaching an agreement, so I carefully read the paper, the review, and the rebuttal. Overall, the weaknesses the reviewers pointed out can be fixed before the camera-ready deadline (as long as the authors do what they promise they will do in the rebuttal). Therefore, I recommend accepting this paper.